# Development of a Low-Cost and High-Efficiency Culture Medium for Bacteriocin Lac-B23 Production by *Lactobacillus plantarum* J23

**DOI:** 10.3390/biology9070171

**Published:** 2020-07-17

**Authors:** Jianming Zhang, Yushan Bu, Chengcheng Zhang, Huaxi Yi, Daqun Liu, Jingkai Jiao

**Affiliations:** 1Institute of Food Science, Zhejiang Academy of Agricultural Sciences, Hangzhou 310016, China; zhangjianming@zaas.ac.cn (J.Z.); zhangcc@zaas.ac.cn (C.Z.); 2College of Food Science and Engineering, Ocean University of China, Qingdao 266100, China; 21170731132@stu.ouc.edu.cn; 3State Key Laboratory of Dairy Biotechnology, Dairy Research Institute, Bright Dairy & Food Co., Ltd., Shanghai 200436, China; jiaojingkai@brightdairy.com

**Keywords:** bacteriocin Lac-B23, low-cost, high-efficiency, culture medium

## Abstract

At present, De Man, Rogosa and Sharpe (MRS) broth is the medium of choice for promoting bacteriocin production. However, this medium is expensive and not applicable for large-scale production. Therefore, a low-cost and high-efficiency culture medium for bacteriocin Lac-B23 production by *Lactobacillus plantarum* J23 was developed. First, the effects of the composition of MRS broth on bacteriocin Lac-B23 production and bacterial growth were researched by a one variable at a time approach. Then, a Plackett-Burman design was used to screen significant components for production. Finally, the steepest ascent and central composite designs were used to obtain an optimum medium. The final composition of the modified MRS was much simpler than MRS broth, and the modified MRS contained only glucose, yeast extract, dipotassium phosphate, manganese sulfate monohydrate, Tween 80 and sodium acetate anhydrous. The highest bacteriocin Lac-B23 production reached 2560 activity units (AU)/mL in the modified MRS, which is nine times higher than that in MRS broth (280 AU/mL). Meanwhile, the cost per liter of the modified MRS (8.56 Ren Min Bi (RMB)/L) is 34.70% the cost of MRS broth (13.11 RMB/L), and the cost per arbitrary units of bacteriocin Lac-B23 in the modified MRS is approximately fourteen times more convenient (3.34 RMB/10^6^ AU) than in the MRS broth (46.82 RMB/10^6^ AU).

## 1. Introduction

Bacteriocins from lactic acid bacteria are bioactive peptides or proteins with antimicrobial activity against different species of Gram-negative and Gram-positive bacteria [1,2,3,4]. Bacteriocins have many advantages, including nontoxicity, no residue, nonresistance, high stability, etc. Therefore, bacteriocins of lactic acid bacteria have been used widely in the field of food preservation as natural food preservatives [5,6]. However, only a few bacteriocins from lactic acid bacteria, such as pediocin and nisin, have been used in commercial applications due to three main reasons: (1) high fermentation costs and low production levels, (2) complex extraction and purification processes and (3) the lack of safety evaluations. Among them, high fermentation costs and low production levels are the biggest bottleneck in the industrial mass production of bacteriocins [7,8,9]. Currently, MRS broth (De Man, Rogosa and Sharpe) is regarded as the beneficial medium for the growth of *enterococcus*, *Lactobacillus* and some bacteriocins synthesis [10,11,12]. However, the components of MRS broth are complex and expensive. Therefore, MRS broth is not suitable for large-scale production [13,14]. The development of a low-cost and high-efficiency culture medium for bacteriocin production is particularly important.

Bacteriocin Lac-B23 is a novel bacteriocin produced by *Lactobacillus plantarum* J23 (*L. plantarum* J23) from Chinese traditional fermented milk and shows antimicrobial activity against *Listeria monocytogenes* (*L. monocytogenes*) [15]. However, the production of bacteriocin Lac-B23 is low in MRS broth. The purpose of this study is to develop a modified MRS (mMRS) culture that can be used for the cultivation of *L. plantarum* J23 and production of bacteriocin Lac-B23. At first, the influence of the composition of MRS broth on bacteriocin Lac-B23 production and bacterial growth was studied by a one variable at a time (OVAT) approach. Then, Plackett-Burman, steepest ascent and central composite designs were used to obtain the best culture medium for bacteriocin Lac-B23 production. The need for developing a low-cost and high-efficiency culture medium for bacteriocin production lies in the final aim of producing a high amount of bacteriocin Lac-B23 to meet the need for further research, such as large-scale purification and research of antibacterial packaging materials. 

## 2. Materials and Methods

### 2.1. Strains and Media

*L. plantarum* J23 was studied as a producer of bacteriocin Lac-B23 and was grown in MRS broth (glucose 20 g/L, peptone 10 g/L, beef extract powder 7.5 g/L, yeast extract 5 g/L, dipotassium phosphate 2 g/L, magnesium sulfate heptahydrate 0.58 g/L, manganese sulfate monohydrate 0.25 g/L, Tween 80 1 mL/L, anhydrous sodium acetate 5 g/L and diammonium hydrogen citrate 2 g/L). *Listeria monocytogenes* (*L. monocytogenes*) CICC 21633, as an indicator bacteria, were grown in a brain heart infusion broth (QingDao Hope Bio-Technology Co., Ltd., Shandon, China). All organisms were propagated twice in their corresponding medium at 37 °C for 24 h before use.

### 2.2. Bacteriocin Assays and Cell Density Measurements

The cell-free supernatants and bacteriocin activity tests were performed by methods published from our laboratory [15]. Briefly, *L. plantarum* J23 was inoculated in fresh MRS at 37 °C for 16h and centrifuged at 8000 rpm for 5 min. The supernatants were adjusted to pH 6.0–6.5 with 1-M NaOH and filtered through a 0.22-µm filter (Millipore Corporation, Billerica, Massachusetts, USA). Bacteriocin activity tests were used by the agar well-diffusion assay. Fifteen millimeters of 1.5% (*w*/*v*) agar were poured into a sterile plate as the bottom medium, and oxford cups of 7.8 mm were put on the above. Molten soft MRS broth agar (0.6%, *w*/*v*) was inoculated with a 10-µL indicator strain (10^8^ CFU/mL). Fifteen milliliters of the inoculated soft agar was poured into the surface of the bottom medium. After solidification, oxford cups were pulled out and 100 µL of the supernatants from *L. plantarum* J23 was placed into a well of 7.8 mm. The plates were incubated at 37 °C, and the zones of growth inhibitions were measured after 12 h. One bacteriocin unit was arbitrarily defined as the reciprocal of the highest dilution, which showed a clear inhibition zone and was expressed as activity units (AU) per mL.

Cell density (OD_600_) was determined at 600 nm using a UV-5100 spectrophotometer (Shanghai Metash Instruments Co., Ltd, Shanghai, China).

### 2.3. Influence of Medium Compositions on Bacteriocin Lac-B23 Production and Bacterial Growth

MRS broth was used as the fermentation medium to research the effects of different compositions on bacteriocin Lac-B23 production and bacterial growth. Experiments were performed using an OVAT approach. To test the influence of the carbon source, various carbon sources (glucose 5–40 g/L, maltose 5–40 g/L, lactose 20 g/L, galactose 20 g/L, fructose 20 g/L, sucrose 20 g/L, xylose 20 g/L and arabinose 20 g/L) were added to the MRS broth without a carbon source to a final concentration of 2%, and the MRS broth without a carbon source was used as a control. To test the effect of the nitrogen source, a modified MRS1 (mMRS1) was chosen as a basal medium (peptone, beef extract powder and yeast extract were omitted). The mMRS1 medium was supplemented with a mixture of multiple nitrogen sources or each individual nitrogen source (Table 1). To study the effects of the dipotassium phosphate concentrations, different concentrations of dipotassium phosphate (0, 1, 2, 5, 7.5, 10, 12.5 and 15 g/L) were added to mMRS2 (mMRS1 was supplemented with 1% yeast extract and without dipotassium phosphate). To investigate the effects of the magnesium sulfate concentrations, different concentrations of magnesium sulfate (0, 0.29, 0.58, 0.87, 1.16, 1.45, 2.03 and 2.5 g/L) were added to mMRS3 (mMRS2 was supplemented with 0.5% dipotassium phosphate and without magnesium sulfate). To study the influences of the manganese sulfate concentrations, different concentrations of manganese sulfate (0, 0.1, 0.25, 0.5, 1 and 2 g/L) were added to mMRS4 (mMRS3 without manganese sulfate). The experiment to determine the effects of the Tween 80 concentrations (0, 0.5, 1, 2, 3, 4 and 5 mL/L) was carried out using mMRS5 (mMRS4 was supplemented with 0.01% manganese sulfate and without Tween 80). To investigate the influences of the sodium acetate concentrations, different concentrations of sodium acetate (0, 2.5, 5, 7.5, 10, 15 and 20 g/L) were added to mMRS6 (mMRS5 was supplemented with 0.3% Tween 80 and without sodium acetate). The experiment to research the influences of the diammonium hydrogen citrate concentrations (0, 1, 2, 5, 7.5, 10 and 15 g/L) was carried out using mMRS7 (mMRS6 was supplemented with 0.75% sodium acetate and without diammonium hydrogen citrate). All experiments were incubated at 37 °C for 16 h. Bacteriocin activity and cell density were measured as described above.

### 2.4. Plackett-Burman (PB) Design

The purpose of this optimization step was to identify which ingredient(s) of mMRS7 had a significant effect on bacteriocin Lac-B23 production. There were six ingredients in the mMRS7 medium, and every ingredient was set as two levels (−1 and +1) (Table 2).

According to the PB design, there were 12 experiments.

### 2.5. Steepest Ascent Experiment

To approach the region of the maximum bacteriocin activity, the next optimization step was carried out along the path of the steepest ascent [16].

### 2.6. Central Composite Design (CCD)

After steepest ascent, the next step was to determine the optimal levels of Tween 80 and sodium acetate. Therefore, the response surface approach by CCD was performed (Table 3).

## 3. Results

### 3.1. Effects of the Composition of MRS Broth on Bacteriocin Lac-B23 Production and Bacterial Growth

The best carbon sources for *L. plantarum* J23 growth were glucose and maltose, followed by fructose, lactose and galactose (Table 1).

In addition, glucose (concentrations from 20 to 35 g/L) or maltose (concentrations from 30 to 35 g/L) as the only carbon sources for growth were also the most suitable carbon sources for bacteriocin Lac-B23 production (280 AU/mL) compared to lactose (160 AU/mL) and galactose (160 AU/mL). Bacterial growth can be improved by adding a yeast extract. Furthermore, the highest production (560 AU/mL) of bacteriocin Lac-B23 was observed when *L. plantarum* J23 was grown in a medium with 10 g/L yeast extract as the only nitrogen source.

In addition, we can conclude that dipotassium phosphate, magnesium sulfate, Tween 80, sodium acetate and diammonium hydrogen citrate do not significantly affect the bacterial growth (Figure 1a,b,d–f). However, manganese sulfate (concentrations above 0.5 g/L) inhibited the bacterial growth. The highest bacteriocin Lac-B23 production (640 AU/mL) was obtained in the presence of 5 g/L dipotassium phosphate, which improved the bacteriocin Lac-B23 production by 15-fold compared with the medium without dipotassium phosphate. This shows that dipotassium phosphate plays an important role in bacteriocin Lac-B23 production. Although magnesium ions participate in protein synthesis, Figure 1b shows that magnesium sulfate is not necessary for bacteriocin Lac-B23 production. Therefore, magnesium sulfate was not added in the follow-up experiments. Bacteriocin Lac-B23 production was enhanced by the addition of manganese sulfate and reached a maximum (800 AU/mL) with 1-g/L manganese sulfate. Tween 80 strongly affected the production of bacteriocin Lac-B23. The optimal production (1120 AU/mL) of bacteriocin Lac-B23 was recorded in the presence of 3-mL/L Tween 80. Moreover, the optimal concentration of sodium acetate required for bacteriocin Lac-B23 production (1280 U/mL) was from 7.5 to 15 g/L, and above or below this concentration decreased bacteriocin Lac-B23 production. The concentration of diammonium hydrogen citrate (from 0 to 10 g/L) had no influence on bacteriocin Lac-B23 production, and high concentrations of diammonium hydrogen citrate (up to 15 g/L) led to a decrease in activity. Hence, diammonium hydrogen citrate was not included in the medium.

### 3.2. Screening of Significant Variables Using Plackett-Burman Design

The Plackett-Burman design is an important technique for screening significant variables and has been used successfully by many researchers [17,18]. According to the results of the OVAT approach, glucose, yeast extract, dipotassium phosphate, manganese sulfate, Tween 80 and sodium acetate played important roles in bacteriocin Lac-B23 production. However, the OVAT approach only studied the influence of a single factor. Therefore, these six variables are analyzed by the Plackett-Burman design to research their synergistic effects. The corresponding response experiments are shown in Table 4.

The results showed that Tween 80 and sodium acetate had the greatest influences on the production of bacteriocin Lac-B23 under synergistic effects. Their contribution rates were 26.32% and 37.69%, respectively. In addition, their *P*-values (*P_Tween 80_* = 0.0194, *P_sodium acetate_* = 0.0136) were lower than 0.05 (shown in Table 5).

This illustrated that Tween 80 and sodium acetate have significant impacts on bacteriocin Lac-B23 production. Therefore, Tween 80 and sodium acetate were selected for further optimization research.

### 3.3. Steepest Ascent Experiment 

To determine the maximum production area of bacteriocin Lac-B23, a steepest ascent experiment was carried out. As shown in Table 6, the concentrations of Tween 80 and sodium acetate increased until experiment 5 and then gradually decreased, implying that the optimal conditions for bacteriocin production are close to the conditions in experiment 5. Therefore, the conditions of experiment 5 were chosen for further optimization.

### 3.4. Central Composite Design Experiment

The result of the CCD experiment for researching the influences of Tween 80 and sodium acetate on bacteriocin Lac-B23 production is shown in Table 7.

The statistical analysis of the results is presented in Table 8.

It shows that the effect of sodium acetate was significant (*p* < 0.05). In addition, the interaction between Tween 80 and sodium acetate, quadratic for Tween 80 and quadratic for sodium acetate, was also significant. The analysis of variance (ANOVA) showed that the *p*-value = 0.0174 and indicated significance for the regression model. The regression equation obtained indicated an R^2^ of 0.9778. This illustrates that 97.78% of the variability in the response could be explained by the model. The following regression Equation (1) was obtained:Y = 24441.212-6017.293*X_5_ − 6430.592*X_6_ + 3986.409*X_5_*X_6_ − 7455.677*X_5_^2^ + 373.33*X_6_^2^ + 903.398*X_5_^2^*X_6_ − 284.44*X_5_*X_6_^2^(1)
where Y is the production of bacteriocin Lac-B23, X_5_ is the Tween 80 concentration and X_6_ is the sodium acetate concentration.

The three-dimensional response surface curves and contour plot were plotted (Figure 2). We obtained the maximum points of the model, which were 1.88 mL/L of Tween 80 and 9.90 g/L of sodium acetate. At this point, the maximum bacteriocin production (2601.78 AU/mL) was achieved. To confirm the predicted results of the model, verification experiments were performed under the predicted conditions. The observed value (2560 AU/mL) was obtained and was reasonably close to the predicted values, confirming the validity of the response model and the existence of an optimal point.

### 3.5. Comparison of MRS Broth and Modified MRS in Terms of Cost and Bacteriocin Lac-B23 Production Level

We obtained a novel modified MRS (mMRS) medium after using the OVAT approach, PB design, steepest ascent experiment and CCD design. As shown in Table 9, when cells were grown in mMRS, the bacteriocin Lac-B23 production increased from 280 AU/mL to 2560 AU/mL compared to the MRS broth. Furthermore, the composition of mMRS is much simpler than the MRS broth, and the cost is drastically lower—from 13.11 RMB/L for the MRS broth to 8.56 RMB/L for the mMRS.

## 4. Discussion

The composition of the medium had a very tremendous influence on the bacteriocin production and bacterial growth. The results indicated that the production of bacteriocin Lac-B23 depends strongly on the type of carbon source. Glucose was the best carbon source for bacteriocin Lac-B23 synthesis. Similar conclusions were described by Schirru [19]. Glucose acts as an inducer and regulates the expression of the genes encoding bacteriocin Lac-B23. However, Zhang et al. (2019) have shown that the rate of plantaricin Q7 biosynthesis in fructose was significantly higher than that in glucose [20]. The main reason for this difference is that the effects of the carbon source on bacteriocin syntheses were related to the strains. For a nitrogen source, a yeast extract (10 g/L) could make the bacteriocin Lac-B23 production reach 560 AU/mL. This implies that a yeast extract contains abundant free amino acids and short peptides for bacteriocin synthesis [11]. However, to some degree, the combination of multiple nitrogen sources was also good for bacteriocin Lac-B23 production. The complex nitrogen source increased the difficulty of isolating and purifying the bacteriocin and increased the cost of the fermentation medium. Therefore, 10-g/L yeast extract was chosen as the only nitrogen source instead of the original organic nitrogen source in the MRS broth. In addition, dipotassium phosphate, manganese sulfate, Tween 80 and sodium acetate had positive effects on the production of bacteriocin Lac-B23. The reason may be that dipotassium phosphate changes the initial pH, or potassium ions are conducive to bacteriocin synthesis. Manganese ions are a constituent of important enzymes involved in glucose metabolism. Therefore, the right amount of manganese ions can provide more ATP from glucose metabolism for bacteriocin Lac-B23 biosynthesis. Tween 80 is an emulsifier that possibly decreases the surface tension of cell membranes. Therefore, some of the metabolites, such as bacteriocins, can be released more smoothly in vitro. Sodium acetate may promote the expression of the bacteriocin Lac-B23 gene as an inducer, or it may play a role in bacteriocin synthesis as a precursor. However, the reason behind the increased bacteriocin production is still unclear. These mechanisms should be studied further.

After the PB design, steepest ascent experiment and CCD design, a novel modified mMRS medium was obtained. The results confirmed that bacteriocin Lac-B23 production can be improved by growing cells in a low-cost mMRS medium. The highest production, 2560 AU/mL, was in a medium consisting of glucose, yeast extract, dipotassium phosphate, manganese sulfate monohydrate, Tween 80 and sodium acetate anhydrous. In particular, the composition of a nitrogen source in the mMRS medium was greatly simplified. Peptone and beef extract powder, as animal origin nitrogens, were completely removed in the mMRS medium. It could reduce untreatable medical conditions, such as Kreutzfeldt-Jacob disease [21]. In addition, a simplified nitrogen source not only facilitated the isolation and purification of bacteriocins but, also, reduced the cost of the medium. Currently, there is interest in the development of a low-cost medium that can be used for the high-efficiency production of bacteriocins [13,21]. In this study, the estimated cost per arbitrary units of bacteriocin Lac-B23 (RMB/10^6^ AU) indicated that bacteriocin Lac-B23 production in the mMRS medium was approximately fourteen times more convenient (3.34 RMB/10^6^ AU) than in the MRS broth (46.82 RMB/10^6^ AU). The possibility of producing increasing amounts of bacteriocin Lac-B23 by *L. plantarum J23* can be considered a challenging opportunity for furthering the range of bacteriocin applications. In particular, bacteriocins can be applied in innovative food packaging. For example, bacteriocin Lac-B23 can be absorbed in a suitable biopolymer and provide an additional barrier, preventing the growth of food-borne pathogens in foods, such as meat, dairy and vegetable products.

## 5. Conclusions

In this study, glucose, yeast extract, dipotassium phosphate, manganese sulfate, Tween 80 and sodium acetate were found to be the key control factors for bacteriocin Lac-B23 synthesis. However, peptone, beef extract powder, magnesium sulfate and diammonium hydrogen citrate were ineffective for bacteriocin Lac-B23 production. Therefore, a low-cost and high-efficiency culture medium for bacteriocin Lac-B23 production was developed. The results show that bacteriocin Lac-B23 production by *L. plantarum* J23 can be improved in a modified MRS culture medium containing glucose, yeast extract, dipotassium phosphate, manganese sulfate monohydrate, Tween 80 and sodium acetate anhydrous. Bacteriocin Lac-B23 production reached 2560 AU/mL in the mMRS, which is nine times higher than that in the MRS broth (280 AU/mL). Meanwhile, the cost per liter of the mMRS (8.56 RMB/L) was 34.70% the cost of the MRS broth (13.11 RMB/L), and the cost per arbitrary units of bacteriocin Lac-B23 in the mMRS was approximately fourteen times more convenient (3.34 RMB/10^6^ AU) than in the MRS broth (46.82 RMB/10^6^ AU).

## Figures and Tables

**Figure 1 biology-09-00171-f001:**
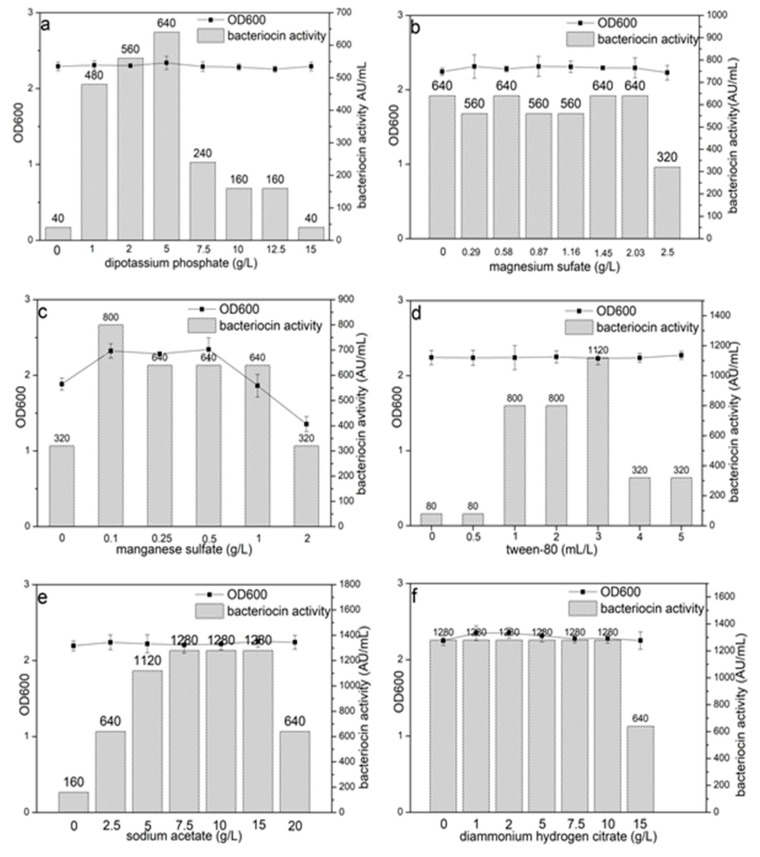
Effects of the ingredients of the De Man, Rogosa and Sharpe (MRS) broth on the growth of *Lactobacillus plantarum* J23 and bacteriocin Lac-B23 production. Note: (**a**) is dipotassium phosphate, (**b**) is magnesium sulfate, (**c**) is manganese sulfate, (**d**) is Tween 80, (**e**) is sodium acetate and (**f**) is diammonium hydrogen citrate. Values listed are the averages of three experiments.

**Figure 2 biology-09-00171-f002:**
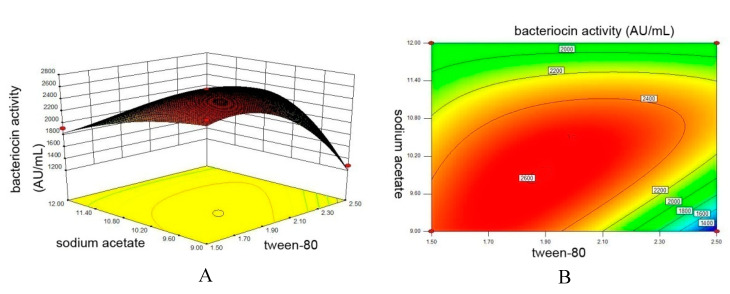
Response surface plot (**A**) and contour plot (**B**) of bacteriocin Lac-B23 production. Values listed are the averages of three experiments.

**Table 1 biology-09-00171-t001:** Influences of the carbon source and nitrogen source on bacteriocin Lac-B23 production and bacterial growth.

Compositions	Concentration g/L	OD_600_ ± SD ^a^	Bacteriocin Activity AU/mL ^a^
control (without carbon source)	0	0.314 ± 0.051	0
arabinose	20	0.260 ± 0.039	0
xylose	20	0.327 ± 0.022	0
sucrose	20	0.635 ± 0.047	20
fructose	20	1.994 ± 0.066	80
galactose	20	2.052 ± 0.066	160
lactose	20	2.042 ± 0.049	160
glucose	5	1.716 ± 0.076	80
glucose	10	2.023 ± 0.106	80
glucose	15	2.193 ± 0.042	240
glucose	20	2.312 ± 0.061	280
glucose	25	2.211 ± 0.069	280
glucose	30	2.260 ± 0.105	280
glucose	35	2.240 ± 0.900	280
glucose	40	2.213 ± 0.109	160
maltose	5	1.820 ± 0.073	160
maltose	10	2.128 ± 0.080	160
maltose	15	2.257 ± 0.065	160
maltose	20	2.294 ± 0.136	240
maltose	25	2.283 ± 0.061	240
maltose	30	2.302 ± 0.047	280
maltose	35	2.275 ± 0.044	280
maltose	40	2.313 ± 0.026	240
control (without nitrogen source)	0	0.214 ± 0.062	0
beef extract powder	7.5	1.918 ± 0.046	40
peptone	10	1.839 ± 0.060	80
yeast extract	1	1.460 ± 0.084	40
yeast extract	2.5	1.989 ± 0.071	160
yeast extract	5	2.223 ± 0.087	280
yeast extract	7.5	2.302 ± 0.034	280
yeast extract	10	2.318 ± 0.051	560
yeast extract	15	2.348 ± 0.093	280
yeast extract	20	2.341 ± 0.029	240
beef extract powder + Yeast extract	7.5 + 5	2.248 ± 0.050	240
Peptone + beef extract powder	10 + 7.5	2.097 ± 0.059	280
Peptone + beef extract	10 + 7.5 + 5	2.263 ± 2.263	280
Peptone + yeast extract	10 + 5	2.236 ± 0.078	320
Peptone + yeast extract	2.5 + 10	2.353 ± 0.052	400
Peptone + yeast extract	5 + 10	2.334 ± 0.078	240
Peptone + yeast extract	7.5 + 10	2.374 ± 0.053	240
Peptone + yeast extract	10 + 10	2.359 ± 0.103	320
Peptone + yeast extract	12.5 + 10	2.337 ± 0.018	400
Peptone + yeast extract	15 + 10	2.343 ± 0.056	480
Peptone + yeast extract	17.5 + 10	2.354 ± 0.055	480
Peptone + yeast extract	20 + 10	2.367 ± 0.085	400

^a^ Values listed are the average of three experiments. SD: standard deviation. AU: activity units.

**Table 2 biology-09-00171-t002:** Range of different factors investigated with Plackett-Burman.

Variable	Real Variables/(Unit)	Low Level (−1)	High Level (+1)
X_1_	glucose/(g/L)	15	25
X_2_	yeast extract/(g/L)	7	13
X_3_	dipotassium phosphate/(g/L)	3	6
X_4_	manganese sulfate/(g/L)	0.08	0.16
X_5_	Tween 80/(mL/L)	2	4
X_6_	sodium acetate/(g/L)	5	10

**Table 3 biology-09-00171-t003:** Range of different factors investigated with the central composite design (CCD).

Variable	Real Variables/(Unit)	Level				
		−1.414	−1	0	+1	+1.414
X_5_	Tween 80/(mL/L)	1.28	1.5	2	2.5	2.71
X_6_	sodium acetate/(g/L)	8.38	9	10.5	12	12.62

**Table 4 biology-09-00171-t004:** Plackett-Burman experiment design and response values.

Run	X_1_	X_2_	X_3_	X_4_	X_5_	X_6_	Bacteriocin Activity (AU/mL) ^a^
1	+1	−1	−1	−1	+1	−1	640
2	−1	+1	+1	−1	+1	+1	960
3	+1	+1	−1	+1	+1	+1	1280
4	+1	−1	+1	+1	+1	−1	640
5	−1	−1	+1	−1	+1	+1	1280
6	+1	+1	+1	−1	−1	−1	1280
7	+1	−1	+1	+1	−1	+1	1280
8	−1	−1	−1	−1	−1	−1	960
9	−1	−1	−1	+1	−1	+1	1920
10	−1	+1	+1	+1	−1	−1	960
11	+1	+1	−1	−1	−1	+1	2560
12	−1	+1	−1	+1	+1	−1	960

^a^ Values listed are the averages of three experiments. X_1_: glucose, X_2_: yeast extract, X_3_: dipotassium phosphate, X_4_: manganese sulfate, X_5_: Tween 80 and X_6_: sodium acetate.

**Table 5 biology-09-00171-t005:** Partial regression coefficients and analyses of their significance.

Factors	Coefficient Estimates	*p*-Values	% Contributions
Intercept	1226.67	0.0465 *	
X_1_	53.33	0.2929	1.05
X_2_	106.67	0.1056	4.21
X_3_	−160.00	0.0513	9.47
X_4_	−53.33	0.2929	1.05
X_5_	−266.67	0.0194 *	26.32
X_6_	320.00	0.0136 *	37.69

* *p*-values less than 0.05 indicated that the model terms are significant. X_1_: glucose, X_2_: yeast extract, X_3_: dipotassium phosphate, X_4_: manganese sulfate, X_5_: Tween 80 and X_6_: sodium acetate. Standard error of all factors is 37.71.

**Table 6 biology-09-00171-t006:** The experimental design and results of the steepest ascent experiment.

Experiment	Tween 80 (mL/L)	Sodium Acetate (g/L)	Bacteriocin Activity (AU/mL) ^a^
1	4	4.5	640
2	3.5	6.0	1280
3	3	7.5	1920
4	2.5	9.0	1920
5	2	10.5	2560
6	1.5	12.0	1920
7	1	13.5	640

^a^ Values listed are the averages of three experiments.

**Table 7 biology-09-00171-t007:** CCD design and response values.

Run	X_5_	X_6_	Bacteriocin Activity (AU/mL) ^a^	
			Expected	Observed
1	0	−1.414	2000	1920
2	+1.414	0	2000	1920
3	0	+1.414	1360	1280
4	−1	−1	2480	2560
5	−1	+1	1840	1920
6	+1	−1	1200	1280
7	0	0	2560	2560
8	−1.414	0	2000	1920
9	0	0	2560	2560
10	0	0	2560	2560
11	+1	+1	1840	1920

^a^ Values listed are the averages of three experiments. X_5_: Tween 80 and X_6_: sodium acetate.

**Table 8 biology-09-00171-t008:** The analysis of variance for the regression model.

Source	Sum of Squares	Degrees of Freedom	*f*-Value	Prob (*p*) > *f*
model	2.257 × 10^6^	7	18.90	0.0174 *
X_5_	0.000	1	0.00	1.0000
X_6_	2.048 × 10^5^	1	12.00	0.0405 *
X_5_ * X_6_	4.096 × 10^5^	1	24.00	0.0163 *
X_5_^2^	4.427 × 10^5^	1	25.94	0.0146 *
X6^2^	1.093 × 10^6^	1	64.06	0.0041 **
X_5_^2^ * X_6_	1.024 × 10^5^	1	6.00	0.0917
X_5_ * X_6_^2^	2.048 × 10^5^	1	12.00	0.0405 *
R-Squared	0.9778			

* *p*-values less than 0.05 indicated that the model terms are significant; ** *p*-values less than 0.01 indicated that the model terms are very significant. X_5_: Tween 80 and X_6_: sodium acetate.

**Table 9 biology-09-00171-t009:** Components, bacteriocin Lac-B23 production and cost analysis for the De Man, Rogosa and Sharpe (MRS) broth and modified MRS (mMRS).

Components ^a^	Price (RMB/g or mL) ^b^	MRS Broth		Modified MRS (mMRS)	
		Concentration (g or mL/L)	Price of MRS Broth (RMB/L)	Concentration (g or mL/L)	Price of mMRS (RMB/L)
glucose	59/500	20	2.36	20	2.36
peptone	139/500	10	2.78	0	0
beef extract powder	299/500	7.5	4.49	0	0
yeast extract	199/500	5	1.99	10	3.98
dipotassium phosphate	79/500	2	0.32	5	0.79
Magnesium sulfate heptahydrate	49/500	0.58	0.06	0	0
manganese sulfate monohydrate	49/500	0.25	0.02	0.1	0.01
Tween 80	119/500	1	0.24	1.88	0.45
Sodium acetate anhydrous	49/500	5	0.49	9.9	0.97
diammonium hydrogen citrate	89/500	2	0.36	0	0
Total medium cost		13.11 RMB/L		8.56 RMB/L	
bacteriocin Lac-B23 production		280 AU/mL		2560 AU/mL	

^a^ All the reagents are AR or BR; ^b^ the price of reagents are based on the Aladdin Industrial Corporation’s price on the Internet (http://www.aladdin-e.com/) in China.

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
