# Peer review of "Development of a Low-Cost and High-Efficiency Culture Medium for Bacteriocin Lac-B23 Production by Lactobacillus plantarum J23"

_biology, 2020, doi:10.3390/biology9070171_

Round 1
Reviewer 1 Report
In this work, the authors report simplified growth medium to obtain enhanced production of bacteriocin Lac-B23 from Lactobacillus plantarum J23. The improved medium contains less components therefore cost-effective, and enhanced the production of bacteriocin Lac-B23, as compared with the traditional MRS medium. The authors' work is important to the field of microbiology. However, I have the following comments that should be addressed.
Specific comments:
1. lines 24-25:
"RMR" should be "RMB"?
2. Lines 54-55, "It was also ----- from other lactic acid bacteria.":
The authors did not use other lactic acid bacteria in this work. Therefore, this part should be removed.
3. Table 1:
How many times were the experiments repeated? The authors should provide the information in the Table footnote and Figure legends.
4. Table 1:
"Power" should be "Powder"?
5. Table 1:
Standard deviations were not shown for bacteriocin activity. It is not clear whether bacteriocin activity were measured for all samples or only one sample per condition. The information should be provided in the footnote.
6.lines 126-127:
This part is based on authors' speculation and does not fit in the result section. Since the authors included this speculation in the discussion section, this part should be removed.
7. Tables 3,4,6,7 and 8:
The readers need to go back to Table 2 to figure out what X1 to X6 indicates. It would be much easier for readers if "real variables"" for X1 to X6 are also included in those Tables or their footnotes.
8. Table 4:
The standard error value (all 37.71) can be included in the footnote of Table 4.
Reviewer 2 Report
The paper by Zhang et al. describes the improvement of a bacteriocin production medium for the lactic acid bacterium strain Lactobacillus plantarum J23. The manuscript holds value for future applications and food-spoilage prevention. It is written in a short and concise way. However, there are some issues.
L14 – MRS broth in not a general medium for bacteriocin production, it’s only applicable for lactobacilli and a handful of lactic acid bacteria that grow on it.
L39 – the same issue as above, only few select species of LAB grow on MRS
L61-62 – L. monocytogenes as inducer bacteria? – Please explain how were they involved.
L66-67 – I think you need to describe the whole procedure.
Ref 10, 11, 19, 21 – bacterial names are written with the first capital letter
Fig. 1 Is a bit of a mess. Place the legend only in fig1A on the right side, then the letters a-f should fit on the left side of each graph.
Reviewer 3 Report
The manuscript written by Jianming Zhang et al. describes a low-cost and high-efficiency culture medium for bacteriocin Lac-B23 production by Lactobacillus plantarum J23. The authors evaluated the components of MRS broth, the most beneficial medium for bacteriocin synthesis by lactic acid bacteria, on bacteriocin Lac-B23 production. The authors described their experimental results in detail (partly little hard to understand), but the impact of this study seems to be limited because it is unclear whether the modified culture could improve the production of other bacteriocins.
There are some discrepancies between the text describing the method and the contents of Table 1. For example, in Table 1, the authors used 5-40 g/L of glucose or maltose, but the text did not. It is unclear whether the authors added various carbon sources into MRS broth or MRS broth without carbon sources (L72-75). It might be necessary to remove either Tween 80, sodium acetate or diammonium hydrogen citrate from each medium to study the effect of the corresponding components. For example, to study the effect of Tween 80, the authors used mMRS5, but it seemed to contain Tween 80. Is it correct?
The manuscript written by Jianming Zhang et al. describes a low-cost and high-efficiency culture medium for bacteriocin Lac-B23 production by Lactobacillus plantarum J23. The authors evaluated the components of MRS broth, the most beneficial medium for bacteriocin synthesis by lactic acid bacteria, on bacteriocin Lac-B23 production. The authors described their experimental results in detail (partly little hard to understand), but the impact of this study seems to be limited because it is unclear whether the modified culture could improve the production of other bacteriocins.
There are some discrepancies between the text describing the method and the contents of Table 1. For example, in Table 1, the authors used 5-40 g/L of glucose or maltose, but the text did not. It is unclear whether the authors added various sources into MRS broth or MRS broth without carbon sources (L72-75). It might be necessary to remove either Tween 80, sodium acetate or diammonium hydrogen citrate from each medium to study the effect of Tween 80, sodium acetate and diammonium hydrogen citrate. For example, mMRS5 seems to contain Tween 80, but it was used to study the effect of Tween 80. Is it correct?
In the first experiment (OVAT approach), the authors showed the growth of lactic acid bacteria and bacteriocin production. But it is also unclear whether the improvement of production of bacteriocin is due to the increase of bacterial growth in the second and third experiments.
The authors discussed the effect of initial pH by adding dipotassium phosphate (L213-214). It may not be difficult to measure the pH of the modified medium.
In the first experiment (OVAT approach), the authors showed the growth of lactic acid bacteria and bacteriocin production. But it is also unclear whether the improvement of production of bacteriocin is due to the increase of bacterial growth in the second and third experiments.
The authors discussed the effect of initial pH by adding dipotassium phosphate (L213-214). It may not be difficult to measure the pH of the modified medium.
Round 2
Reviewer 3 Report
I have just sent a message to the editor.
Author Response
Dear reviewer,
Thank you very much for your comments about our manuscript. we will continbue to refine our reasearch.